# A Review on Recycling of Carbon Fibres: Methods to Reinforce and Expected Fibre Composite Degradations

**DOI:** 10.3390/ma15144991

**Published:** 2022-07-18

**Authors:** Amiruddin Isa, Norlin Nosbi, Mokhtar Che Ismail, Hazizan Md Akil, Wan Fahmin Faiz Wan Ali, Mohd Firdaus Omar

**Affiliations:** 1Department of Mechanical Engineering, Centre for Corrosion Research (CCR), Institute of Contaminant Management for Oil and Gas (ICM), Universiti Teknologi PETRONAS, Bandar Seri Iskandar 32610, Malaysia; amiruddin_20001971@utp.edu.my (A.I.); mokhtis@utp.edu.my (M.C.I.); 2School of Materials and Mineral Resources Engineering, Engineering Campus, Universiti Sains Malaysia, Nibong Tebal 14300, Malaysia; hazizan@usm.my; 3School of Mechanical Engineering, Faculty of Engineering, Universiti Teknologi Malaysia, Skudai 81310, Malaysia; wan_fahmin@utm.my; 4Faculty of Chemical Engineering Technology, Centre of Excellent Geopolymer & Green Technology (CEGeoGTech), Universiti Malaysia Perlis (UniMAP), Kangar 01000, Malaysia; firdausomar@unimap.edu.my

**Keywords:** recycling, carbon fibre, fibres, environmental degradation

## Abstract

Carbon fibres are widely used in modern industrial applications as they are high-strength, light in weight and more reliable than other materials. The increase in the usage of carbon fibres has led to the production of a significant amount of waste. This has become a global issue because valuable carbon fibre waste ends up in landfill. A few initiatives have been undertaken by several researchers to recycle carbon fibre waste; however, the properties of this recycled material are expected to be worse than those of virgin carbon fibre. The incorporation of polymers, nanoparticles and other hybrid materials could enhance the overall properties of recycled carbon fibre waste. However, the degradation of fibre composites is expected to occur when the material is exposed to certain conditions and environments. The study of fibre composite degradation is crucial to enhance their properties, strength, safety and durability for future applications.

## 1. Introduction

Carbon fibres are made up of long chains of carbon atoms that are fused together. The diameter of carbon fibre is approximately 5 to 10 micrometres, which is smaller than the diameter of a strand of human hair [1]. Carbon fibres are usually used in premium products because of their high price as compared to other types of fibre such as glass fibres and plastic fibres. Carbon fibres are super stiff and high-strength, but are extremely light in weight. Thus, they are widely used as building materials as well as in transportation and high-technology industries. There are a variety of carbon fibre structures or shapes, namely weaves, braids, “raw” building blocks and yarns. Carbon fibres are generally applied to reinforce composite materials due to their high strength-to-weight ratio. By comparison, their properties are quite similar to steel and their weight is similar to plastic [2,3,4].

Due to several specific properties possessed by carbon fibres, they are being utilised in numerous industrial applications, products and technologies. Carbon fibres have high tensile strength and toughness with low relative weight. Therefore, they have been used in the manufacturing of the body of aerospace and marine vehicles as well as sports equipment [5,6]. A total of 50% of the Boeing 787 Dreamliner and Airbus A350 aeroplane bodies are made up of carbon fibres [7]. Many cyclists prefer purchasing carbon fibre bicycles for tournaments or daily usage due to their light weight. Carbon fibres have a low thermal expansion and abrasion coefficient, while being highly rated for dimensional stability [8]. Due to these advantages, carbon fibres are the chosen material in the production of missiles, aeroplane antennas, gigantic telescopes and waveguides, especially for measuring high-frequency precision frames with stability. Almost all of the high-quality audio equipment and robotic arms in use require materials with excellent vibration damping, high strength and toughness [9]. Therefore, carbon fibres are the best choice of raw material to manufacture these products. Due to their good electrical conductivity, carbon fibres can be utilised to create electromagnetic interference (EMI) or radiofrequency (RF) shielding as well as electronic device bases [10,11]. Carbon fibres possess good electromagnetic properties, which are the reason for their application in the production of radiological medical equipment. In addition, they are also utilised in building retaining rings for large generators.

In addition, carbon fibres have a unique characteristic, in that they do not respond to or react with biological tissues, which means that the material is biologically inert [12]. Thus, the host will not be harmed if any carbon fibre material is inserted into its biological tissues. Apart from that, the material is also X-ray-permeable. Thus, X-rays and gamma rays can pass through without interference. Researchers combine these two characteristics to develop carbon fibre medical implants, surgery and X-ray utilities, and prosthesis for missing parts of the body. Furthermore, a huge part of today’s engineering success is due to discovery of carbon fibre materials with self-lubricating properties and a high resistance to fatigue [13,14]. Self-lubrication is caused by the self-lubricating (SL) tribolayer effect of carbon fibre incorporated in the matrix, which has an opposite effect of an alloy sliding surface. Hence, a large amount industrial machinery is built using carbon fibres, including textile industrial machinery. Carbon fibres also have good chemical inertness [15] and high corrosion resistance. These features led to the utilization of carbon fibres in the chemical industry. For example, certain devices or machine parts that are always exposed to chemicals, such as valves, components in the pumps, seals and other few elements in the processing plants are manufactured from carbon fibres.

## 2. Demand and Wastage of Carbon Fibre

The high potential and unique characteristics possessed by carbon fibres have led to a significant increment in their market demands. The carbon fibre market is growing bigger and bigger, starting from the last decade, as the result of great utilization and application in various industrial sectors. According to a carbon fibre market report in 2019, the market is expected to expand from USD 4.7 billion to USD 13.3 billion in the ten years starting from 2019 [16]. A carbon fibre compound annual growth rate (CAGR) of 12.5% is projected between 2019 and 2029 [17]. The largest segments recorded for its carbon fibre demands are in the aerospace and defence end-use industrial sectors. This is the consequence of the growth of aircraft production by Boeing and Airbus. It is forecasted that Boeing and Airbus will request approximately 9000 newly built wide-body aircraft in the period of the next 20 years. Similarly, no decreasing trend is observed in carbon fibre usage for aircraft with a narrow body. As stated by The Teal Group, an annual growth rate of 5.8% is expected in the aircraft production market size (USD 164 billion up to USD 218 billion from 2017 to 2022) [18].

Based on illustration data in Figure 1, the increment of global carbon fibre demands throughout the year can be seen. The increase in the global demand for carbon fibre affects the amount of carbon fibre waste. It is difficult to reduce and address the gap between carbon fibre demand and supply. In a report by Composites World, approximately 30% of carbon fibre production ends up being waste [19]. However, the great news is that most analysts estimate that annual carbon fibre demand has the potential to surpass the present capacity of annual production in the next few years. As stated by Brett Schneider, the president at Hexcel (a carbon fibre manufacturer), during the Carbon Fibre Conference held in December 2018, the actual carbon fibre annual production capacity is approximately 150,000 metric tonnes, while the expected global demand for carbon fibre is within the range of 65,000 to 85,000 metric tonnes annually [20]. As reported by Amanda Jacob (Composites World contributor) in March 2019, the opposite trend is expected to occur by a few analysts, in which the carbon fibre annual demand will exceed the supply by 24,000 metric tonnes in the year 2022, but this may take time [21].

Despite this great news, the excessive reduction of carbon fibres into waste has become the main issue and has triggered environmental concerns. The combination of carbon fibre with plastic polymer resin resulted in a strong and light composite, which is preferred by material industries. However, during the manufacturing processes, the material is often laid up by hand and almost one-third of the actual size of carbon fibre sheets is left unused after being trimmed, as stated by the ELG Carbon Fibre company [22]. As a result, a huge amount of the carbon fibre does not end up becoming a product, but rather is sent to landfill. A staggering amount of approximately 62,000 tonnes of unutilised end-of-life (EoL) carbon fibre waste are accumulated each year, with the aircraft and wind energy industries contributing the most to this amount. If no actions are taken, by 2035, a predicted cumulative amount of 23,600 tonnes of unused EoL carbon fibre waste will be produced by the aircraft sector, with an additional 483,000 tonnes from the wind turbine industry. The total accumulation of carbon fibre waste produced from these two sectors shocks the world [23].

In response to the global issues of the effect of carbon fibre waste on the environment, several legislative methods have been implemented. For example, the European Union (EU) introduced a policy to limit carbon fibre waste disposal while encouraging the usage of waste as alternative resources. A five-step waste hierarchy was constructed to deal with the waste, with the top step being avoiding waste, followed by reusing and recycling excess carbon fibre residue, and then by the various methods of recovery, and lastly waste disposal if each of the four steps that are previously mentioned are unable to be implemented [24]. Other than that, restriction of landfilling and incineration as a part of disposal methods for carbon fibre waste is introduced, aiming to promote the recycling of the material. Carbon fibre composite recycling technologies were invented a few years ago, and yet these recycling industries are still young and new. The carbon fibre recycling initiative is still in its early stages to introduce and develop recycled material for new industrial products. Recycling carbon fibre is seen as a potential resolution to minimise the gaps formed between carbon fibre supply and demand, but it does have a few challenges. One of the challenges is that carbon fibre cannot simply be melted and reshaped using moulds, much like aluminium. Hence, the aim of this review paper is to explore possible ways of recycling carbon fibre, provide methods of reinforcement, as it is commonly known that the recycled material will tend to have worse properties overall, and finally to discuss expected degradation studies on this material so that its safety can be ensured before being used in specific industrial applications.

## 3. Recycling of Carbon Fibre

There is no argument that carbon fibre is environmentally friendly and exhibits a longer life cycle. However, carbon fibre consumes almost 14 times more energy in its creation compared with steel. This significant energy-intensiveness has led to huge emissions of greenhouse gases. Therefore, the recycling process could be one of the best ways to reduce this environmental impact while meeting global demand for this material in industrial applications. Currently, carbon fibre waste or other fibre composites can be recycled using four types of technologies. There are two main types of carbon fibre waste. The first type of waste is virgin carbon fibre offcuts of the product generated from dry fibre and the non-used expired material, which are also called scrap. The second type of waste is the reclamation of fibres from carbon fibre-reinforced composites (CFRC). Figure 2 shows all types of recycling processes for both scrap and the composite type of carbon fibre waste.

The first type of method is mechanical-based recycling processes. This process consists of breaking down the carbon fibre waste or carbon fibre-reinforced polymer composites into tiny pieces by crushing, shredding, milling or other similar breaking methods. The scrap produced is then strained and categorised into products with a high content of powdered resin (fine recyclate) and products with a high fibrous content (coarse recyclate). This process does not release poisonous or hazardous gas. The size of the final recycled products can vary in the range of 50 µm to 10 mm [26]. However, the limitation of mechanical-based recycling is that it could only be implemented as a partial reinforcement material for other composites or as a charge due to limited incorporation with base material [27]. Thus, the process is suitable for composite wastes with known origins and are relatively uncontaminated. Fine recyclate has the potential to be reused in the application of the original polymer matrix, while for coarse recyclate, the reinforcement properties are not as good as those of the virgin fibre. Hence, the degradation studies for recovered fibres are crucial because the re-manufacturing of the inconsistent and non-constructed coarse recyclate is quite difficult [28]. A continuation of this journal review topic led to a research project. The research project was carried out and funded by the same group and financial institution as this publication. Figure 3 shows recent progress in recycled carbon fibre composite study samples under this project funded by the Ministry of Education Malaysia.

The microstructure of recycled carbon fibre waste for this project was examined under a scanning electron microscope (SEM). Figure 4 shows the SEM images of recycled carbon fibre waste after milling.

Some of fibre surfaces and cross-sectional shapes were retained after milling. A portion of the recycled carbon fibre lost its fibrous shape and broken.

The second method of recycling is by using a thermal process, which is also known as pyrolysis or oxidation. This is the most widely used method of recycling carbon fibre-reinforced polymer composites. During this process, organic molecules in carbon fibre-reinforced polymer are decomposed through a heating process in a chemically inactive atmosphere. The commonly implemented atmospheres are vacuum, superheated steam and nitrogen gas. The temperature for the thermal decomposition is set within the range of 350 to 700 °C [29,30,31]. A great achievement was accomplished by Meyer et al., where the properties of the recycled carbon fibre were almost at the same level as those of virgin carbon fibre after the pyrolysis process that was set up with optimum parameters via technical viability [32]. Through the degradation of the polymer matrix into smaller molecules, relatively pure carbon fibres were able to be retrieved. In a pyrolysis process that was carried out by Giorgini et al., approximately all polymer matrices were degraded. After the recycling process, the non-affected carbon fibre could be retrieved as well as a fraction of oil and gas, which could be used as chemical feedstock and a source of energy [33]. Unfortunately, this process was quite dangerous, as several hazardous gases were released, and there was a possibility of char marks (burn marks) being left on the surface of the carbon fibre. The formation of char could be completely avoided by designing and applying a commercial semi-closed nonstop belt furnace with a controlled atmosphere. Another type of thermal recycling process is through the fluidised-bed procedure, which is conducted by balancing the downward gravitational force on the particles with the upward force generated by a high flow of gas. This process involves polymer matrix decomposition together with the release and reclamation of discrete carbon fibres. The optimum processing temperatures are carefully determined so that they are suitable for polymer decomposition while leaving clean and non-degraded carbon fibres reclamation afterwards that usually are damaged by extreme temperature exposure [34]. Generally, the carbon fibres that are retrieved from this process are in the form of short fluffy fibres, which could be used to reinforce metal matrix composites such as aluminium alloy matrix composites.

The other type of recycling process for carbon fibre-reinforced polymer is performed by utilizing chemical reactions. Solvolysis is one of the methods used to break down the resin and retrieve non-defective carbon fibres, fillers and depolymerised polymer matrix in two forms, namely petrochemical feedstock and monomers [35]. Solvolysis also tolerates waste with a low level of contamination such as non-metal-based material or material with painting application. The solvolysis process can be customised by changing the temperature, the applied pressure and the addition of solvents and catalysts. A few reactive media such as catalytic solutions, subcritical and supercritical fluids, benzyl alcohol and many more are used generally at low temperatures. The medium is selected based on the polymer substrate nature, which has already been mechanically shaped to enhance the total surface area. There are two categories of solvolysis, namely the high pressure and temperature (more than 200 °C) category and the low pressure and temperature (lower than 200 °C) category. Relatively large-size oligomers (repeated polymer molecule units) could be obtained from the decomposition of polymeric resin, as well as inactive carbon fibre. Approximately 90% of virgin carbon fibre mechanical properties can be restored through the chemical recycling process. However, high reagent costs and massive emissions of harmful gases become a huge challenge for this type of process. Thus, environmentally friendly water or alcohol typically replace highly concentrated and dangerous chemicals [36].

Xu et al. measured tensile strength of a single recycled carbon fibre, which was almost 95% that of virgin carbon fibre. The chemical solution used in the study was a mixture of N, N-dimethylformamide and hydrogen peroxide (H_2_O_2_). Images taken using a scanning electron microscope (SEM) showed that the carbon fibre surface was relatively smooth with little epoxy resin residue left on it [37]. Degradation by oxidation at 60 °C for a period of 30 min resulted in clean carbon fibres, the tensile strength of which was 95% higher than the original strength of the material. The chemicals used in the study were acetone solution together with hydrogen peroxide [38]. Apart from that, the use of subcritical and supercritical fluids or benzyl alcohol as a catalyst could recover semi-long and long carbon fibres through the process of resin rapid breakdown. However, the recycling process of subcritical and supercritical fluids requires high-value parameters and conditions. Specifically for carbon fibre-reinforced composites, solvolysis using subcritical and supercritical fluids as solvents requires temperatures higher than 300 °C and a pressure above 50 bar, whereas the composite recycling process could be performed at ambient conditions. Meanwhile, the application of catalysts, such as acidic solutions or strong alkaline solutions, can accelerate the chemical reaction at lower temperatures, although it can damage the fibres. Mixtures of acidic solutions are hazardous to humans and the environment. Therefore, this process demands high-value reactors, which can withstand extreme temperatures and pressures as well as possess high corrosion resistance. In conclusion, there is a need to evaluate the pros and cons of using these high-cost facilities and valuable solvents before they can be utilised at an industrial scale.

The last method for recycling carbon fibre is the size-reduction method using high-voltage, electrohydraulic and electrodynamic fragmentation. High-voltage fragmentation (HVF) was conducted using a high-voltage pulse of approximately 160 kV to break down fibre-reinforced polymer. The result of this process was long and clean fibres [39]. However, research conducted by Oshima et al. resulted in a significant mass loss in the composites and a reduction in resin removal rate after the implementation of high-voltage fragmentation [40]. Another method is electrodynamic fragmentation (EDF), which can be performed by applying a high-voltage pulse in the range of 50 to 200 kV to ionised water to break down the carbon fibre-reinforced polymer waste into tiny pieces [41,42]. In addition, electrohydraulic fragmentation (EHF) also can separate carbon fibre-reinforced polymer waste into carbon fibres and the polymer matrix. Recycled carbon fibres that are retrieved from this process are ready to be used because of their high quality, and there is almost no polymer matrix left on the fibres’ surface [43]. The process involves strong sound wave pulses, which separate fibres from the polymer resin when they hit the material. Table 1 lists a few recent studies on the carbon fibre recycling process and the extraction of carbon fibre from composites.

## 4. Methods to Reinforce Recycled Carbon Fibres Properties

It was observed that the overall properties of post-recycled carbon fibres experience a reduction in quality due to the shortening of average fibre length. However, this can be altered by reinforcing the post-recycled carbon fibres with polymer matrix and certain hybrid materials and nanoparticles to achieve better overall properties. Although the carbon fibres are recycled or extracted from carbon fibre-reinforced polymer composites, their behaviour and chemical reaction are observed to be similar to those of virgin carbon fibres. Due to the lack of research and studies carried out by researchers on recycled carbon fibre properties, the virgin carbon fibre properties could be examined and set as a benchmark for the study of recycled carbon fibres, the properties of which are expected to be worse in terms of overall strength as compared to the virgin version.

In research conducted by Alsaadi et al. in 2018 on the effects of nano-silica inclusion on the mechanical behaviour of carbon fibre with epoxy resin hybrid composites, it was found out that the inclusion of 3 wt% nano-silica particles resulted in an improvement in tensile strength by 20% and by 35.7% in material flexural strength [50]. Another study was performed by Singh et al. in 2017, in which silicon dioxide nanoparticles were included in epoxy polymer nanocomposites by adding 4 wt% silica using ultrasonication. The ultimate tensile strength increased by 30.57%, the material flexural strength by 17% and the flexural modulus elasticity by 76% from their respective original values. However, the addition of silica of more than 4 wt% resulted in a reduction in the ultimate tensile strength of the specimen. This might be due to the larger cluster size because of the increase in nanoparticle loading in the matrix, hence reducing the bound clusters’ strength, which led to cracking [51]. Suresh et al. studied the effect of the addition of nanofillers such as titanium oxide (TiO_2_) and silicon carbide (SiC) on the mechanical properties of epoxy hybrid composites. Three nanofiller weight percentages, namely 0, 5 and 10%, were tested on the specimen using the hand layup method of manufacturing. The optimum nanofiller content was approximately 5 wt%. The addition of nanofillers beyond 5 wt% would reduce the resulting tensile, flexural and impact strength, together with its hardness [52].

In research conducted by Dass et al. in 2015 on the post-mechanical properties of chopped carbon fibre-reinforced polymer composites after the addition of alumina nanoparticles, it was found that the tensile strength of the material increased from 48–58.54 MPa to 96–110 MPa with only 6 wt% carbon fibre contents and 3 wt% alumina nanoparticles. The same trend was observed in the flexural and compression strength of the specimen, where the values increased from 115–156.56 MPa to 176–204.66 MPa and 48–61.15 MPa to 72–85.65 MPa, respectively. The mechanical properties of the material started to exhibit opposite trends when more nanoparticles were added [53]. In another published research paper by Mohanty et al. (2014), it was discovered that a concentration of 2 wt% alumina nanoparticles was the optimum value to enhance the thermal stability and flexural modulus and strength as well as the impact strength of short carbon fibre-reinforced polymer composites. The combination of 2 wt% alumina nanoparticles and 5 wt% short carbon fibre with epoxy resulted in a 130% improvement in flexural strength and a 55% increment in the flexural modulus [54].

A review paper written by Zakaria et al. on hybrid carbon fibre with carbon nanotube-reinforced polymer composites revealed that the mechanical properties of the material were enhanced as a result of an effective interface, which acted as a bridge between the hybrid material (carbon fibre–carbon nanotubes) and polymer. The hybrid material helped in reducing the stress concentration due to its unique three-dimensional network structure. Apart from that, a comparison was made between numerous mechanical properties of carbon fibre–carbon nanotubes polymer composites that were fabricated using various methods [55]. Moaseri et al. (2014) reported a significant enhancement in tensile strength and the tensile modulus of hybrid carbon fibre–carbon nanotubes polymer composites by 126% and 233%, respectively, through the electrophoretic deposition method of fabrication. The stiffening effect provided by the carbon nanotube coating on the carbon fibre surface was achieved through the interlocking of the polymer matrix on top of the fibre surface area. The polymer used in the study was epoxy [56]. Another similar hybrid material that was tested by Rahmanian et al. with similar fabrication methods also showed improvements in tensile strength and tensile modulus, where the matrix used was polypropylene (PP). Improvements by 57% and 50% from the original tensile strength and tensile modulus, respectively, were caused by the dispersion of carbon fibre in the polypropylene matrix that led to higher strength [57]. A change was made by Wang et al. (2018) in the matrix of a polymer by using phenolic compounds, which resulted in an increase in tensile strength by approximately 45% compared to that of the raw material [58].

## 5. Degradation of Recycled Carbon Fibre Composites

Based on the literature, fibres undergo degradation when the material is exposed to certain types of conditions and environments. Due to certain applications of carbon fibre composites which expose them to high surrounding temperature (such as gun barrel, bridges, retrofitting of metallic building), seawater (as base material for construction of yacht, canoe and boat) and acidic and alkaline environments (used as reinforcement for damaged round steel pressure pipelines in which mostly used as economical transmission of liquids and gases), it is important to study the degradation of the material in this sector. Researchers need to study the degradation of recycled carbon fibre composite before it can be used as a material for industrial appliances. However, there are only a limited number of studies on recycled carbon fibre composites, and so far, no degradation studies have been carried out which can be used as a reference. Table 2 lists the studies on the reduction in the quality of the general properties experienced by fibres and fibre-reinforced polymer composites when they are exposed to degradation conditions and environments. These studies are used as a benchmark for this work.

### 5.1. Effects of Moisture Absorption

In a matrix-based degradation, the degrading factors penetrate through the structure of the composites. Thus, stress corrosion of reinforcement fibres will begin and trigger the acceleration of size delamination propagation. This results in the destruction of the element due to stability loss. In the modern chemical recycling of carbon fibre, the degradation condition can be achieved by using solvents through solvolysis or water through hydrolysis. The solvolysis process is implemented to break down or depolymerise the composite polymeric parts in several degradation processes, consisting of solvents at various concentrations and within various reactions. On the other hand, degradation occurs due to water replacement in the material during the hydrolysis process [70].

In general, the degradation mechanism caused by moisture absorption requires a certain period to be completed. Each specimen is weighed at a specific time interval. Equation (1) is implemented to evaluate the percentage of moisture absorbed:(1)$$M_{t}\left(\%\right)=\frac{W_{t}-W_{i}}{W_{i}}\times 100$$
where *M_t_* represents the weight of moisture absorbed by the specimen as a percentage, *W_t_* is the weight of the specimen at a specific time interval, and *W_i_* is the initial weight of the specimen.

The coefficient of moisture diffusion into the specimen, *D*, can be used to analyse moisture absorption. *D* is expressed as follows:(2)$$D=\pi \cdot {\left(\frac{kh}{4\cdot M_{m}}\right)}^{2}$$
where *M_m_* represents the maximum weight gained after moisture absorption, *k* is the initial slope gradient of graph *M* (%) against *t*^1/2^, and *h* is the thickness of the specimen [71]. This plot graph is crucial as confirmation that the specimen has achieved effective equilibrium of moisture. Whether it is necessary to make a judgement of the engineering and to study the behaviour of the material moisture properties for the case where effective equilibrium is not reached can be decided using the graph plot.

Water absorption in fibre-reinforced polymer composites occurs through three main mechanisms, which are capillary, diffusion and water molecules transportation. The diffusion mechanism occurs inside the micro gaps that exist between polymer chains. The mechanism of transportation in the capillary occurs in the holes that form in the interface space of the fibre matrix [72,73]. The swelling in the fibre composites caused by water absorption allows water molecules to be transported through the microcracks that evolve in the fibre matrix. All of these complex phenomena, such as plasticization of the matrix, relaxation, swelling, interfacial debonding in the matrix or fibre, and rearrangement of the chemical structure, are affected by the exposure of the material to certain environments [74]. The diffusion of water molecules from the fibre shell layer to the carbon fibre core is due to the different concentrations of two-phase media until the concentration becomes equal throughout.

According to Huo et al. in summer 2016, the most common fundamental used by researchers to study the effects of moisture diffusion into fibre-reinforced polymer composites is referring to Ficks’s one-dimensional law. Unfortunately, this classical law is not always the best choice to explain moisture diffusion [74]. The absorption of moisture into polymer composites could potentially reduce the quality of the initial physical and mechanical properties, thus altering the overall composite performance. In addition, the degradation of the polymeric matrix could occur when exposed to corrosive substances, and this decreases the tensile strength. The polymers’ molecules will be disturbed by corrosive chemical substances or other moisture sources. Even ultraviolet rays may affect the strength of interfacial bonds in polymer composites [75]. Drzal et al. pointed out that the degradation of carbon fibre epoxy has reversible and irreversible effects depending on the temperature of hygrothermal exposure, the application of mechanical stress and the type of treatment applied to the surface of the fibre [76]. Wang et al. proved that the combination of electrochemical oxidation and treatment on carbon sizing potentially enhanced the interfacial shear strength (IFSS) from 59 to 70%. The specimen was subjected to a microbond pulling test after being exposed to a relative humidity of 95% at 40 °C for 3 weeks and 3 days [77]. Cauich-Cupul et al. studied the effect of moisture on carbon fibre epoxy composites specifically in bulk resin together with the interphase of a fibre–polymer matrix. The result of the fragmentation test on a single fibre revealed that the uptake of moisture by the matrix, which caused swelling to occur, resulted in the extreme deterioration of the interfacial shear properties caused by residual stresses. After radial stresses were drastically reduced, the mechanical components of the fibre-polymer matrix adhesion also lessened [78].

Glaskova-Kuzmina et al. discovered that the addition of multi-walled carbon nanotubes in carbon fibre reinforced plastic reduced the coefficient of moisture absorption by 31% and equilibrium content of moisture by 15%, and improved creep resistance in the short-term cyclic creep-recovery test. The study also revealed that the reduction in the flexural modulus was approximately by a multiple of 1.4 and 3 [79]. Bond et al. computationally explored the relationship between the spatial distribution of fibre and diffusivity of moisture. The results of the study show that the coefficient of diffusion was not related to the random distribution of fibre compare to the regular version [80]. Aditya et al. discovered that the diffusivity of moisture was affected by the fibre cross-sectional shape. The study concluded that the circular fibres possessed a minimum level of moisture diffusivity, and that this would increase when the circular-cross section eccentricity increased [81].

### 5.2. Effects of Ageing

From an investigation conducted by previous researchers, the ageing of fibre composites may have effects on the mechanical properties of the material. According to the study by Li et al., the tensile strength and modulus of elasticity decreased over time to a certain value. They remained constant before later recovering slightly. This might be due to the equilibrium between water and the material, where no water molecules diffused into and out of the matrix and the fibre matrix interface. In the initial 40 days of immersion, the tensile strength showed a decreasing trend. However, from day 40 to 120, the ultimate strength showed a slight increment in its value. A similar trend was also recorded for the elastic modulus of the specimen after it was left in the seawater for approximately 120 days. Within 63 days, there is a total reduction in the modulus of 9 GPa, while the value illustrates a slight increase in the elastic modulus by 6 GPa by the end of immersion period [82].

In 2009, Menail et al. studied the effect of water ageing on composites’ properties after 100, 500 and 1000 h of immersion. Both types of fibre, Kevlar and glass fibre, experienced a reduction in ultimate stress and strain as the immersion time increased. The behaviour of the material remained quasilinear before rupture occurred [83]. Phifer et al. discovered that the tensile strength and stiffness of fibre-reinforced polymer composite reduced by approximately 60% and 10%, respectively, after long immersion in fresh water lasting approximately 2 years [84]. Jiang et al. observed a reduction in the mass of fibre-reinforced polymer composites starting at an ageing condition of 40 °C. A humid ageing environment supported by elevation in temperature increased the rate of moisture diffusion and lessened the time required to obtain an initial equilibrium of moisture [85]. Lee et al. found that the increase in thermal ageing of fibre-reinforced natural rubber composite resulted in oxidation and cross-linking reactions in the rubber matrix. The oxygen content of the natural rubber matrix increased with an increase in thermal ageing time and temperature because of the oxidation that occurred on the surface as the temperature rose [86]. Figure 5 illustrates the oxidation and breaking of chains caused by the ageing process on bio-based fibre composites.

Eslami et al. carried out a study on the relationship between ageing temperature and moisture absorption in fibre-reinforced polymer composites. The study revealed that an elevation in ageing temperature reduced the coefficient of diffusion, hence inducing a higher level of absorption of moisture by the composite, which in turn caused a reduction in the flexural stiffness and bending capacity of the composite [87].

### 5.3. Effects of Exposure to High-Temperature Environments

Several researchers confirmed that changes in the temperature, moisture absorption and duration of moisture exposure can alter the overall performance of polymer composites. According to small considerations in theoretical thermodynamics, solubility is affected by temperature. According to the equation derived by Van’t Hoff, the dependency of solubility on temperature is determined by the change in enthalpy as a consequence of dissolution [88].
(3)$$\frac{d\ln c_{s}}{dT}=\frac{∆H\left(T\right)}{RT^{2}}$$
where *c_s_* is the saturation concentration, ∆*H* is the dissolution heat, *T* is the temperature in Kelvin, and *R* is the gas constant. Through interpretation, the absorption of water into a polymer can be referred to as a dilute solution, and it is predicted to follow the equation derived by Van’t Hoff. As an alternative to dissolution heat, the determination of the temperature dependency of saturation weight gain is achieved by calculating the absorption heat. For the measurement of normal water absorption within a small temperature range, the absorption heat is assumed to be constant, thus integrating Equation (3) to produce:(4)$$c_{s}=c_{s,ref}\cdot \exp\left[-\frac{∆H_{abs}}{R}\left(\frac{1}{T}-\frac{1}{T_{ref}}\right)\right]=c_{s,ref}\cdot \exp\left(-\frac{∆H_{abs}}{RT}\right)\cdot \exp\left(\frac{∆H_{abs}}{RT_{ref}}\right)\;$$
define
(5)$$c_{s0}=c_{s,ref}\cdot \exp\;\left(\frac{∆H_{abs}}{RT_{ref}}\right)$$
then
(6)$$c_{s}=c_{0}\cdot \exp(\frac{-∆H_{abs}}{RT})\;or\;M_{\infty }\left(T\right)=M_{0}\cdot \exp\left(\frac{-∆H_{abs}}{RT}\right)$$
where *M_∞_* (*T*) is the equilibrium water uptake as a function of temperature, ∆*H_abs_* is the absorption heat per mole of absorbed water molecules, and *M*_0_ is the temperature-independent constant. Referring to Equations (4) to (6) above, in an endothermic absorption process, an increment in temperature results in an increase in equilibrium uptake, whereas the equilibrium uptake decreases with the rise in temperature in exothermic absorption [89]. The absorption heat is affected by interactions between the polymer matrix and the absorbent. The negative absorption heat may be caused by the reaction between the polar polymer and moisture. In detail, the absorption of moisture was reported to be a slightly exothermic, which caused the epoxy level of saturation to reduce with an elevation in temperature. Meanwhile, if there are no chemical reactions between water and the polymer matrix, the moisture absorption heat is expected to be small. Thus, the temperature will have a minor effect on equilibrium uptake.

In research by Hailin et al., carbon-reinforced polymer composite specimens submerged in artificial seawater at high temperatures experienced extreme degradation, and changing the sodium chloride concentration of the seawater did not affect the tensile strength [90]. Another study was carried out by Ibrahim et al. to investigate the effect of temperature on glass fibre epoxy composite joints. Three specimens were immersed in hot water at three different temperatures for certain periods. It could be seen from the results that the specimens immersed in hot water at 50 °C absorbed 0.31 wt% moisture, and 0.71 wt% moisture was absorbed by the specimen immersed in 70 °C hot water. At 90 °C, the specimen absorbed 0.76 wt% moisture. Overall, the high absorption of moisture by the specimen was due to resin matrix damage that occurred as the temperature of the water increased. The weakening in the matrix and interfacial fibre matrix led to larger gaps, and this allowed the water to be absorbed into the composite plate. Post-immersion inspection of the specimen revealed that the higher temperature of water, the weaker the joints, hence decreasing the tensile strength [91]. Hagihara et al. in 2018 stated that nylon6-based carbon fibre-reinforced thermoplastics (mixture of epoxy resin, polyamide and polycarbonate) degraded after being exposed to high-temperature and high-pressure steam. The molecular bond strength of the material decreased, as well as its bending strength from 811 to 255 MPa. Another study was conducted in 2020 by Li et al., in which it was discovered that the tensile strength together with the residual thermal stress of E-glass fibre polymers decreased with increasing exposure temperature [66]. Figure 6 shows an illustration on how the delamination process occurred after the polymer matrix was broken down due to the rise in fluid temperature.

Wan et al. observed the decrement in time taken to reach saturation level and the rate of absorption when the temperature of medium increased [92]. This may be caused by the severe extensive corrosion of the fibre composite surface and its interior caused by temperature, thus loosening the interface of the carbon fibre matrix [93]. Gabrion et al. investigated the tensile properties of carbon fibre-reinforced high-temperature thermoplastic composite after being exposed to high temperatures. They concluded that thermal degradation occurred at temperatures higher than 400 °C [94]. The interlaminar shear strength was relatively low when the material was exposed to high temperatures. Vielle et al. stated that quasi-isotropic carbon fibre composite experienced a decrement of 3% in the modulus of elasticity, 8% in overall strength and 5% in overall strain as the temperature increased from ambient to 120 °C [95]. Uematsu et al. detected a decrease in the tensile rigidity by approximately 7% when the temperature rose from room temperature to 200 °C [96]. Mahieux et al. observed a reduction in the cyclic failure number for unidirectional AS4/PEEK samples when the temperature was set to 150 °C [97,98].

### 5.4. Effects of Seawater Exposure

Seawater may influence the bending strength of reinforced carbon hybrid fabric. This can be seen in the experiment conducted by Komorek et al. in 2016, in which the samples that were exposed to seawater experienced a 23% decrement in bending strength on average. A similar trend was observed in the samples’ flexural strength after being immersed in seawater, where the flexural strength decreased by 6%. Immersion in seawater caused the fracture mechanism to change from cracking of the matrix to failure at the interface. The swelling process that occurred during immersion may be associated with the effects of bonded water, in which crucial internal stresses were exerted by the presence of a constraint [99]. In a similar study by Li et al. (2016) on the effect of seawater on reinforced carbon woven fabrics, similar effects were observed on the material properties. The tensile strength of the specimen after being immersed in seawater experienced a reduction from 1415 to 1190 MPa. The modulus of elasticity of the specimen also declined in value from 106 to 97 GPa after being immersed in seawater for approximately 63 days [82].

In 2016, Sun et al. immersed a carbon fibre-reinforced polymer anode strip in sodium hydroxide (NaOH) solution and pore water solution while applying a current of 0, 0.5 and 4 mA. The evident final degradation rate of the strip anode with an applied current of 4 mA was 12.4 μm/day for the NaOH solution and 13.6 μm/day for the pore water solution [100]. A few other studies, in which the degradation properties of various types of fibre composites were mainly tested after being immersed in various salt solutions, showed similar trends in the reduction in tensile strength. Wang et el. found that hybrid basalt-carbon fibre-reinforced polymer tendons experienced degradation under high stress in a salt solution [65]. Similar results obtained by researchers proved that fibre composites degraded when they were exposed to moist conditions with a high salt content [59,60,62]. Yuan et al. implemented the quantum dots analysis technique to study the degradation of a fibre-reinforced polymer composite after being immersed in seawater. After 6 months of immersion, a decrease of 2.5% in glass transition temperature, 13.8% in ultimate tensile strength and 9.8% in maximum flexural strength was observed [62]. Figure 7 shows an illustration of the quantum dots technique used to study the breakdown of interfacial bonding between the polymer resin and glass fibre in this study.

### 5.5. Effects of Acidic Solution Exposure

An analysis performed by Arya et al. in 2020 revealed that carbon fibre-reinforced polymer composites absorbed more moisture when the material was exposed to hydrochloric acid (HCl) [101]. High contents of hydrochloric acid diffused into the cracks that existed in the specimen. The polymeric matrices that were analysed using scanning electron microscopy (SEM) were damaged and the degradation process was believed to occur. The tensile strength of specimen that was immersed in hydrochloric acid decreased by 24.77% from the original value due to fibre damage, interphase matrix debonding and produced voids. Furthermore, prolonged immersion in hydrochloric acid at elevated temperatures potentially accelerated the degradation process, and its rate could be predicted using the Arrhenius equation:(7)$$k=A\exp\left(-\frac{E_{a}}{RT}\right)$$
where *k* is the rate of degradation, *A* is a constant (differs for each material), *E_a_* is the activation energy, *R* is the universal gas constant and *T* is the temperature of the solution in Kelvin. In the Arrhenius equation, it is assumed that the mechanism of degradation is not affected by time and solution temperature. However, the degradation rate increases as the temperature increases. Equation (7) can be simplified into Equations (8) and (9):(8)$$\frac{1}{k}=\frac{1}{A}\exp\left(-\frac{E_{a}}{RT}\right)$$
(9)$$\ln\left(\frac{1}{k}\right)=\frac{E_{a}}{RT}\cdot \frac{1}{T}-\ln A$$

Equation (8) represents the degradation rate, where *k* is the inversely proportional to the time needed for the material properties to achieve a certain value. Equation (9) is the natural logarithm of the time needed by the material properties to achieve a certain value, and it is a linear function of 1/*T*. The gradient of the slope is equal to *E_a_*/*RT*.

Ji et al. concluded that sulfuric acid activated carbon fibre-reinforced polymer aliphatic amine and alkane functional groups as well as chain scission reactions, which deteriorated the interfacial bonding between resin and carbon fibres. The energy storage, the capacity of dissipation and glass transition temperature of carbon fibre-reinforced polymer composite sheets experienced a reduction in their values by 10% after being exposed to sulfuric acid for 6 weeks. Nevertheless, carbon fibres were not directly damaged because they were shielded by the epoxy resin. The high deterioration rate of the epoxy interface softened the material strain, thus reducing the maximum load that can be carried by the composites [102]. The effect of an acidic solution on fibre-reinforced plastic pipe was investigated by Mahmod et al. at ambient and high temperatures. After 30 days of immersion in 20% hydrochloric acid at an ambient temperature, no effect on the flexural strength of the composite was observed. However, for a longer immersion time, the flexural strength of the composite reduced by approximately 10% [103]. Kanerva et al. conducted the long-term immersion of a fibre-reinforced polymer composite in a sulfuric acid solution for about 24 weeks at 90 °C and 15 bar. The results show that the acidic solution significantly reduced the tensile strength by approximately 6% to 49% and the flexural stiffness by approximately 13% to 34% [104]. Cousin et al. concluded that an E-glass fibre-reinforced polymer bar had the least acid resistance, as the fibre experienced the highest weight loss of approximately 21.9% to 35.1% after being immersed in hydrochloric acid [105].

### 5.6. Effects of Alkaline Solution Exposure

Judd stated in his research that carbon fibres are inert to chemicals such as alkaline solutions at any concentration or temperature (up to boiling point). However, they will experience degradation when they are combined with a polymer matrix [106]. The coupling agents that make up chemical bonds in carbon fibre epoxy may be ruptured due to the high concentration of hydroxyls that come into contact with them. An example of the most popular coupling agent is silane, which is commonly added to glass fibres. Elevation in temperature of the alkaline solution will further accelerate the degradation process, as stated by Benmokrane et al. [107]. Several studies have suggested the idea of combining glass fibres with carbon fibres as composites due to their potential strength enhancements. However, glass fibres are subject to damage when alkaline environments are introduced. The fibre breakdown is a combination of two processes: first, the alkaline solution attacks the glass fibre and second, the hydration products grow in between individual fibre filaments [108]. Calcium hydroxide nucleation on the surface of the fibre resulted in fibre embrittlement. The hydroxylation process forced the surface of the fibre to become rough and pitting to occur. Aggressive ions such as calcium, potassium and sodium could accelerate the degradation when combined with other types of alkali salts [109]. Figure 8 demonstrates how aggressive ions diffuse into microcracks and result in fibre breakdown.

In a study by Katsuki et al., electron probe microscopy was utilised to track the diffusion of alkali ions into various types of fibre-reinforced vinyl ester rods. From their observation, the radial direction penetration of sodium ions into the glass fibre-reinforced rods started as soon as the samples were immersed in the alkaline solution. However, after 60 days of immersion in alkaline solution, no degradation was detected for both the carbon fibre and amide-reinforced polymer samples [110]. In an experiment that was carried out by Almusallam et al., glass fibre-reinforced polymer bars were immersed for 18 months in an alkaline solution at 50 °C. The tensile properties of the samples decreased by 1 to 2% due to degradation. Scanning electron microscopy (SEM) was utilised to study the mechanism of degradation (water diffusion and alkali ions into matrix) in the study [111].

## 6. Conclusions

The potential for the future utilization of recycled carbon fibre in large industrial-scale applications is very high. Modern fibre recovery technologies and research works are still in progress to ensure that the characteristics of recycled carbon fibre are almost identical to those of virgin carbon fibre. At present, carbon fibre is very useful and widely used in today’s application because of its light weight, which is similar to plastics, while it is super stiff and strong. Thus, numerous recycling methods have been proposed and additional materials have been added into recycled carbon fibre to produce high-quality composites, taking advantage of the interfacial bonding between fibres and the matrix that has been explored by researchers. However, only a few degradation tests have been carried out to study their mechanical, thermal and chemical properties.

The understanding of the degradation processes and the degrading conditions is crucial in determining the recycled carbon fibres’ endurance, their important properties and their safety before being commercially used in industrial applications. Several scholars have discovered that the fibre properties such as mechanical strength, fatigue life, residual thermal stress and many more decreased after the material experienced degradation. Therefore, similar degradation tests should be carried out on recycled carbon fibre to explore its true potential for implementation in various future applications.

## 7. Future Suggestions

The deteriorated properties of recycled carbon fibre could be enhanced by mixing the carbon fibre with other suitable materials. In this process, the composites should undergo a degradation process and be exposed to degrading conditions for a certain period, and their properties should be analysed afterwards. For this purpose, suitable experimental works need to be designed to monitor the behaviour of the material post-degradation. The mechanical recycling process should be chosen, as this process is considered the safest choice at the moment. A suitable polymer matrix and additional hybrid material should enhance the overall properties of recycled carbon fibre polymer composites after the fibres are crushed, shredded or milled for recycling purposes. The recycled carbon fibre polymer composites should be exposed to various degrading conditions and solutions. Mechanical, thermal and chemical tests should be performed on the composites after the completion of the degradation process, along with imaging of the microstructure as well as results analysis. In this way, the material’s reliability could be determined, along with whether it is suitable for future industrial applications or whether an improvement of its overall properties is required.

## Figures and Tables

**Figure 1 materials-15-04991-f001:**
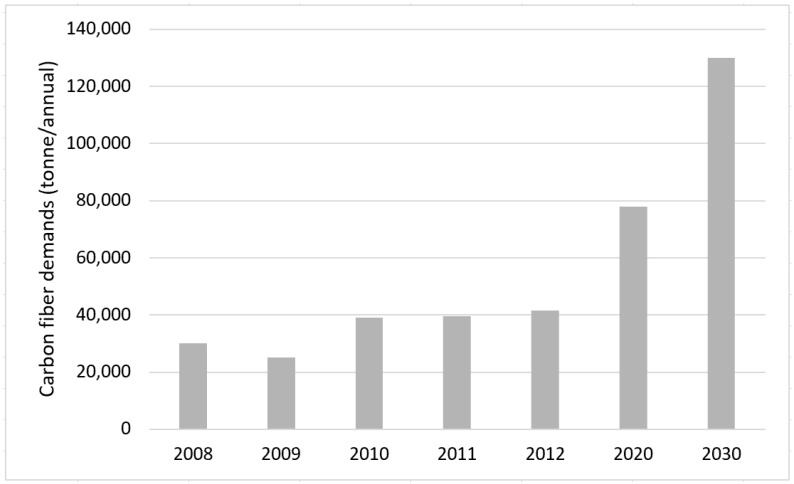
Increase in carbon fibre global demand.

**Figure 2 materials-15-04991-f002:**
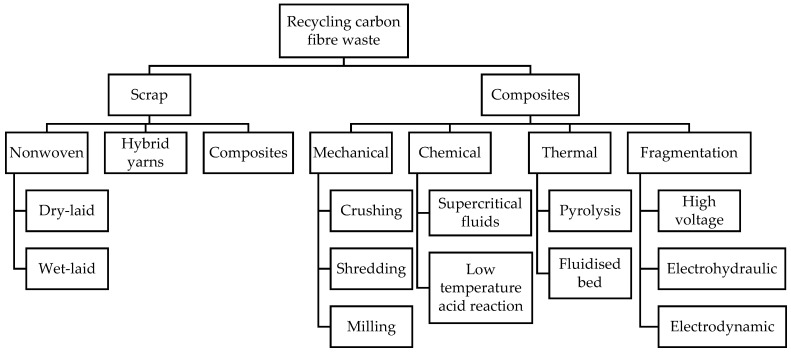
Methods of recycling carbon fibre waste [25].

**Figure 3 materials-15-04991-f003:**
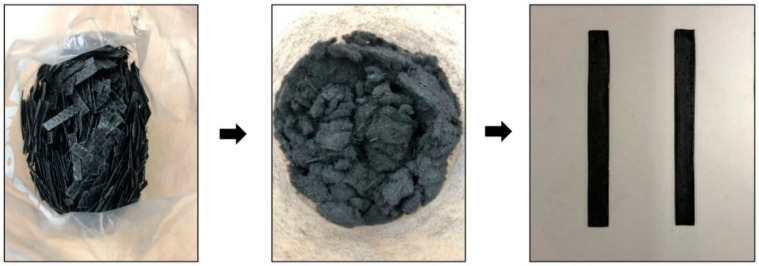
Example of recycled carbon fibre composite samples by mechanical process (pictures were originally taken at the Centre of Corrosion Research, Universiti Teknologi Pertronas).

**Figure 4 materials-15-04991-f004:**
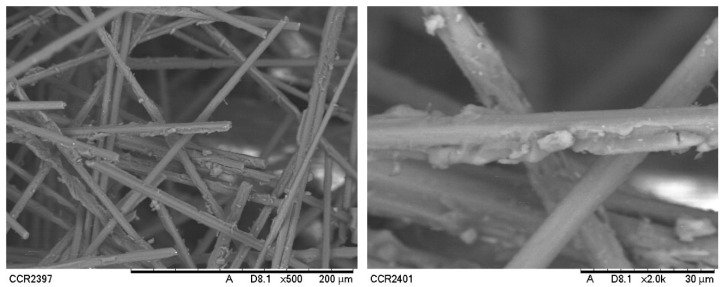
SEM images of recycled carbon fibre waste with ×500 and ×2000 magnification (pictures were originally taken at the Centre of Corrosion Research, Universiti Teknologi Pertronas).

**Figure 5 materials-15-04991-f005:**
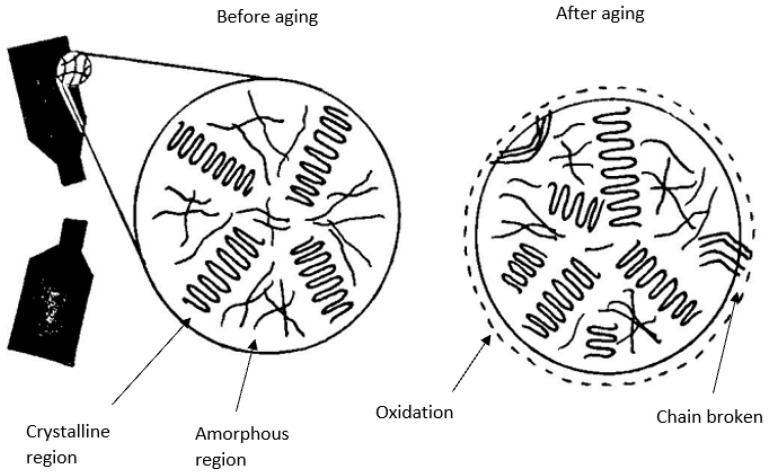
The effect of ageing on bio-based fibre composites.

**Figure 6 materials-15-04991-f006:**
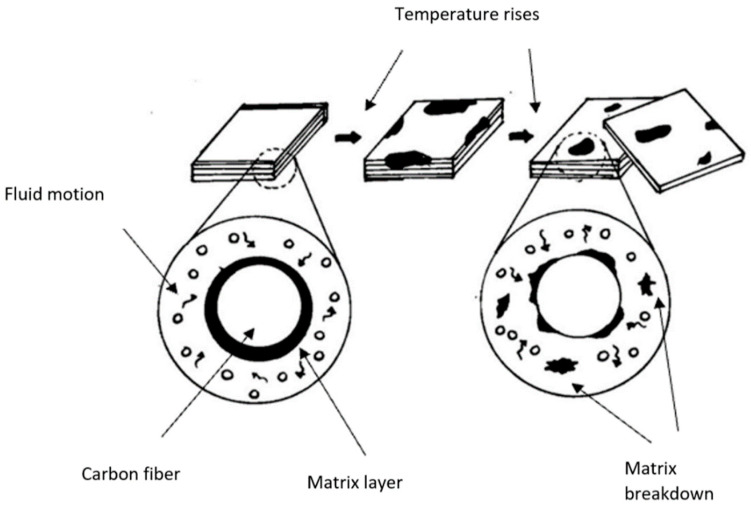
Breakdown of polymer matrix due to elevation in fluid temperature.

**Figure 7 materials-15-04991-f007:**
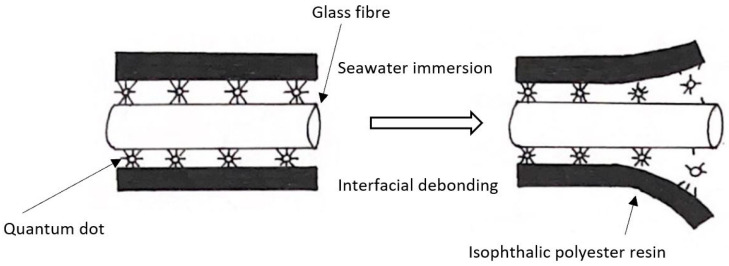
An illustration of the quantum dots analysis technique for the study of fibre-reinforced polymer composite degradation in seawater.

**Figure 8 materials-15-04991-f008:**
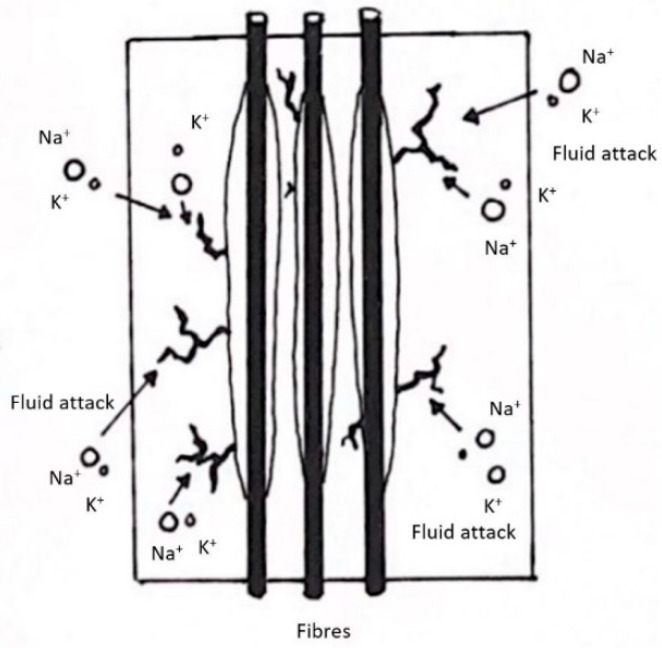
Aggressive ions attack microcracks in the fibre.

**Table 1 materials-15-04991-t001:** Recent research on the extraction and recycling of carbon fibres.

Year	Type of Material and Product	Recycling Ways	Parameters	Result	Ref.
2014	Carbon fibre-reinforced plastic (CFRP) → carbon fibre + thermoplastic resin	Chemically (selective decomposition)	Supercritical methanol at 270 °C at 8 MPa	Carbon fibre tensile strength reduced by 9%	[44]
				Carbon fibre interfacial strength reduced by 20%	
2020	Carbon fibre-reinforced polymer composites (CFRP) waste → carbon fibre	Thermally (two-step thermolysis)	30 min of pyrolysis at 500 °C and 40 min of oxidation	93.47% carbon fibre recovered	[45]
2020	Recycled carbon fibres → recycled carbon fibre-reinforced polyamide 6 (nylon 6) and polypropylene	Mechanically and thermally (using simulation of injection moulding on Moldflow)	Injection pressure: 50 MPa to 150 MPa	Tensile strength increased 427% for nylon 6	[46]
				Tensile strength increased 781% for polypropylene	
2021	Recycled carbon fibres → recycled carbon fibre-reinforced polypropylene	Mechanically and thermally (using injection moulding)	Fibre weight percentage: 10, 20 and 30 wt%	Stiffness and strength increased linearly with weight percentage of fibre	[47]
				18.5% improvement in tensile strength (30 wt%)	
2021	Unsaturated polyester resins (UPRs) → carbon fibre	Chemically (via hydrolysis-oxidation synergistic catalytic strategy)	80% N_2_H was added into NaOH	Recycled carbon fibre tensile strength was 3.05 GPa	[48]
				Virgin carbon fibre tensile strength was 3.33 GPa	
2021	Recycled carded nonwoven carbon fibre → recycled discontinuous carbon fibre organosheet	Thermally (compression moulding)	Heated press at 300 °C under 1 MPa pressure for 5 min	Recycled fibre type samples recorded 94 MPa for tensile strength compared to the virgin version at 139 MPa.	[49]

**Table 2 materials-15-04991-t002:** Recent research on the degradation of fibre-reinforced composites.

Year	Type of Material	Type of Degradation	Parameters	Result	Reference
2010	Phosphate glass fibre-reinforced methacrylate-modified oligolactide	Salt solution Tris buffer solution	0.9 wt% NaCl solution at 37 °C Tris buffer solution	pH dropped from 7.26 to 7.11 Tensile strength decreased with the addition of calcium carbonate	[59]
2013	Natural fibre-reinforced polymer matrix composites (using sawdust and wheat flour for fibre and polypropylene for matrix)	Moist soil Water 10% salt solution Sunlight	Material exposed to degradation solution for 15 weeks	Tensile strength decreased from 32.1 MPa to: 29 MPa for wheat flour 26 MPa for sawdust	[60]
2016	Carbon fibre-reinforced polymer strip anode	NaOH solution Simulated pore water solutions	Applied current ampere: 0 mA 0.5 mA 4 mA	Degradation rates at 4 mA were: 12.4 μm/day for NaOH 13.6 μm/day for pore water solution	[61]
2017	Glass fibre-reinforced polymer using isophthalic polyester resin	Seawater	Immersed in artificial seawater and deionised water for 6 months	Glass transition temperature decreased by 2.5% Tensile strength decreased by 13.8% Flexural strength decreased by 9.8%	[62]
2018	Nylon 6-based carbon fibre-reinforced thermoplastics (CFRTP) using epoxy resin, polyamide and polycarbonate	Hot steam (at various temperatures)	Temperature and pressure: 120 °C at 198.7 kPa 140 °C at 361.5 kPa 120 °C at 618.2 kPa	Molecular weight decreased Bending strength decreased	[63]
2019	Toray T700 6 k carbon fibre fabric	Acetone	Temperature of 300 °C for 1 h	Resin removal yield (RRY): 31.8% without alkaline and weak Lewis acid 24.4% and 40.3% with alkaline and weak Lewis acid	[64]
2019	Basalt fibre-reinforced epoxy composites	Saltwater	Mass of salt in 1 L of water at 25 °C: 24.53 g of NaCl 4.09 g of Na_2_SO_4_ 5.2 g of MgCl_2_ 1.16 g of CaCl_2_	Negligible reduction in static strength (less than 5%) Fatigue life decreased with ageing time	[65]
2020	E-glass fibre with epoxy resin, E-glass fibre with polyester resin, E-glass fibre with vinyl ester resin and glass fibre with polypropylene	Temperature	Temperature dependence of Weibull stress modulus and fibre strength at a certain gauge length was negligible	Tensile strength decreased with an increase in temperature Residual thermal stress decreased with an increase in temperature	[66]
2020	*Calotropis gigantea* plant fibre-reinforced polymer composites	Water	Immersion conducted for 72 h	Absorption rate increased with an increase in fibre content	[67]
2020	Carbon fibre reinforced polycarbonate	Hot water	Deionised water at 80 °C	Tensile properties decreased rapidly with time after 7 days	[68]
2021	Virgin and pyrolysed carbon fibre	Thermal (using thermogravimetric analysis	Heating rate at 5 and 10 K/min	Pyrolysed carbon fibre degraded at lower temperatures compared to virgin carbon fibre	[69]

## Data Availability

Not applicable.

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
