# Peer review of "A Review on Recycling of Carbon Fibres: Methods to Reinforce and Expected Fibre Composite Degradations"

_materials, 2022, doi:10.3390/ma15144991_

Round 1

Reviewer 1 Report

The review "A Review on Recycling of Carbon Fibres: Methods to Reinforce and Expected Fibre Composite Degradations"  shows an very important topic neglected from past research. There only some minor points to address and some suggestions

1. The authors speak about environmental issues of carbon fibers in landfills. It would be beneficial and helpful to define which are those as example what is the toxicity of carbon fiber in environment. Please include a short section about such so reader understand why recycling is needed. 

2. Figure 3 and Figure 4 looks like from actual experiments. There are no references given where pictures taken and if from other work please include permission of such.

Minor issues. There several parts in text as example space between sentences (line 216 as example) Please check manuscript throughout. Another part please check all units some have space other don't (line 406).

Author Response

Norlin Nosbi

Mechanical Engineering Department,

Universiti Teknologi PETRONAS,

32610 Seri Iskandar

Perak, Malaysia

Tel: +605 368 7145

Email: norlin.nosbi@utp.edu.my

June 26, 2022

Dear Dr.,

Thank you for giving me the opportunity to submit a revised draft of my manuscript entitled “A Review on Recycling of Carbon Fibres: Methods to Reinforce and Expected Fibre Composite Degradations” to Materials. We appreciate the time and effort that you as reviewer have dedicated to providing your valuable feedback on our manuscript. We are grateful for your insightful comments on our paper. We have been able to incorporate changes to reflect most of the suggestions provided by you. We have highlighted the changes within the manuscript.

Comments from Reviewer 1

  • Comment 1: The authors speak about environmental issues of carbon fibers in landfills. It would be beneficial and helpful to define which are those as example what is the toxicity of carbon fiber in environment. Please include a short section about such so reader understand why recycling is needed.

Response: We agree with your thought of this matter. Therefore, we already explained one and the only discovered environmental impact of carbon fiber. (from line 134 to 139 in section 3). The explanation as per below.

“There is no argument carbon fiber is environmentally friendly and exhibits longer life of cycle. However, carbon fiber consumes almost 14 times more energy for its creation to be compare with steel. This significant amount of energy-intensive has led to huge omission of greenhouse gases. Therefore, recycling process could be one of the best way to reduce this environmental impact while meeting its global demand in industrial applications.”

  • Comment 2: Figure 3 and Figure 4 looks like from actual experiments. There are no references given where pictures taken and if from other work please include permission of such.

Response: Thank you for pointing this out. Actually, Figure 3 and 4 are originally from the our research project in recycling the carbon fiber waste but it never been published to any journal yet as the project is still on going. This research project is part of continuation from this review publication. A description of the project is explained from line 158 to 160.

  • Comment 3: Minor issues. There several parts in text as example space between sentences (line 216 as example) Please check manuscript throughout. Another part please check all units some have space other don't (line 406).

Response: Thank you for pointing this out. All the space between issues have been corrected.

Sincerely,

Norlin Nosbi

Reviewer 2 Report

The reviewed manuscript entitled: A review on recycling of carbon fibres: Methods to reinforce and expected fibre composite degradation is written welll and clear, however the the aim of this whole review is missing. What was the goal? just to inform the reader about possible recycling methods?

Some other comments:

1.      Figure 2 is confusing. Is scrap and composite on the same level? Is scrap not obtained until after the mechanical recycling process?

2.      Page 4, Line 154

Figure 3 shows the recent progress of recycled carbon fiber composite study samples under this project funded by Ministry of Education Malaysia.   

... this project ... means this publication? If the authors mean some research project, they could describe it. Also, is Figure 3 original or already published somewhere? The reference is missing if it was published already—the same for Figure 4.

3.      Page 5, lines 188-190

Generally, the carbon fibers retrieved from this process are in the form of short fluffy fibre, which could be used to treat metal-based contaminated waste. 

Do the authors mean that fluffy fiber could play a role in reinforcement in metal composites? Or could you be more focused?

4.      Is the chemical and thermal recycling route equally effective for all types of polymer matrices? Could the authors also discuss the influence of the type of polymer matrix on the choice of recycling way?

5.      On the other hand, does the material from which carbon fibers are made affect the properties of carbon composites and, later, the properties of composites after recycling? Are there studies that look at differences in recycling routes, whether have carbon fibers been prepared from the pitch, polyacrylonitrile, or cellulose? Discussion on this topic is entirely missing. Can they comment on the impact of the input material on the recycling path and the properties of the recycled material?

6.      If Table 1 lists recent research on the extraction and recycling of carbon fibers, then the column Fabrication means the recycling way?

7.      In my opinion, the title: Methods to Reinforce Recycled Carbon Fibres Properties, is not suitable. The methods described in that part 4 improved the properties of composites reinforced with carbon fibers.

8.      Could you briefly discuss the applications related to the effects of seawater, high temperature, alkaline or acidic environments? Why is it essential to observe the impact of the mentioned environments on the properties of carbon fiber composites? Could such composites be used in ships? What application is related to the alkaline and acidic environments?

Author Response

Norlin Nosbi

Mechanical Engineering Department,

Universiti Teknologi PETRONAS,

32610 Seri Iskandar

Perak, Malaysia

Tel: +605 368 7145

Email: norlin.nosbi@utp.edu.my

June 26, 2022

Dear Dr.

Thank you for giving us the opportunity to submit a revised draft of my manuscript entitled “A Review on Recycling of Carbon Fibres: Methods to Reinforce and Expected Fibre Composite Degradations” to Materials. We appreciate the time and effort that you as reviewer have dedicated to providing your valuable feedback on our manuscript. We are grateful for your insightful comments on our paper. We have been able to incorporate changes to reflect most of the suggestions provided by you. We have highlighted the changes within the manuscript.

Comments from Reviewer 2

  • Comment 1: The reviewed manuscript entitled: A review on recycling of carbon fibres: Methods to reinforce and expected fibre composite degradation is written well and clear, however the the aim of this whole review is missing. What was the goal? just to inform the reader about possible recycling methods?

Response: Thank you for your response for the aim of this review paper. The aim of this review paper is to explore possible ways of recycling carbon fiber, methods of reinforcement in which it is commonly known recycled material will tend to have lower in overall properties and the last is to discuss general ideas expected degradation studies on this materials so that its safety been ensured before being used in specific industrial applications. Line 132 to 136 already added in manuscript to give the focus/aim of this review paper publication.

  • Comment 2: Figure 2 is confusing. Is scrap and composite on the same level? Is scrap not obtained until after the mechanical recycling process?

Response: You have raised an important point here. Therefore, we already added short line 140 to 143 in section 3 to explain about it. The sentences as per below:

“There are two main types of carbon fiber waste. First, virgin carbon fiber offcuts from product generated from dry fibre which also called as scrap. The second type of waste is reclamation of fibre from carbon fiber-reinforced composites (CFRC).”

  • Comment 3: Page 4, Line 154

Figure 3 shows the recent progress of recycled carbon fiber composite study samples under this project funded by Ministry of Education Malaysia.  

... this project ... means this publication? If the authors mean some research project, they could describe it. Also, is Figure 3 original or already published somewhere? The reference is missing if it was published already—the same for Figure 4.

Response: Thank you for pointing this out. Actually, Figure 3 and 4 are originally from the our research project in recycling the carbon fiber waste but it never been published to any journal yet as the project is still on going. This research project is part of continuation from this journal review paper publication. A description of the project is explained from line 158 to 160 in section 3.

  • Comment 4: Page 5, lines 188-190

Generally, the carbon fibers retrieved from this process are in the form of short fluffy fibre, which could be used to treat metal-based contaminated waste.

Do the authors mean that fluffy fiber could play a role in reinforcement in metal composites? Or could you be more focused?

Response: Agree. We have revised and changed the sentence (line 203 to 206 in section 3) to emphasize this point. Short carbon fiber are previously known to be used as a reinforcement for metal matrix composites such as aluminium alloy matrix composites.

  • Comment 5: Is the chemical and thermal recycling route equally effective for all types of polymer matrices? Could the authors also discuss the influence of the type of polymer matrix on the choice of recycling way?

Response: Thank you for your suggestion. It would have been interesting to explore this aspect. However, due to limited of research paper on recycling for specific inclusion of polymer matrix in carbon fiber, it is a little bit difficult to discuss about the effectiveness and suggestion of recycling ways. It also seems slightly out of our focus on carbon fiber itself. Our scope of this publication is to discuss general ideas on ways of recycling for the other upcoming research and projects in the future to be investigated. Due to similariton of virgin carbon fiber and carbon fiber composites chemical reaction and properties, it has been used as a benchmark for this review.

  • Comment 6: On the other hand, does the material from which carbon fibers are made affect the properties of carbon composites and, later, the properties of composites after recycling? Are there studies that look at differences in recycling routes, whether have carbon fibers been prepared from the pitch, polyacrylonitrile, or cellulose? Discussion on this topic is entirely missing. Can they comment on the impact of the input material on the recycling path and the properties of the recycled material?

Response: Thank you for your suggestion. As similar to previous comment, due to limited of research papers on this recycling matter, plus with the process of recycling carbon fiber is still in early phase, information about inclusion of material and post effects by recycling of carbon fiber with different creation sources are still lacks and narrowly discovered.

  • Comment 7: If Table 1 lists recent research on the extraction and recycling of carbon fibers, then the column Fabrication means the recycling way?

Response: Thank you for pointing this matter. We believe suggestions of using “Recycling Way” for “Fabrication” column would be more appropriate. Thus, change have been made for “Fabrication” column in Table 1.

  • Comment 8: In my opinion, the title: Methods to Reinforce Recycled Carbon Fibres Properties, is not suitable. The methods described in that part 4 improved the properties of composites reinforced with carbon fibers.

Response: You have mentioned an important point here. However, we believe studies discussed in section 4 focus more on reinforcement towards carbon fiber or carbon fiber composites because all the researchers used carbon fiber or carbon fiber composites as their base material. For example, from line 272 in section 4, Alsaadi et el added nano-silica  to improve carbon fiber epoxy composites tensile strength. Another example discussed in line 303, section 4, Zakaria et el revealed hybrid material which is carbon nanotubes acted as a bridge and helped in reducing the stress concentration.

  • Comment 9: Could you briefly discuss the applications related to the effects of seawater, high temperature, alkaline or acidic environments? Why is it essential to observe the impact of the mentioned environments on the properties of carbon fiber composites? Could such composites be used in ships? What application is related to the alkaline and acidic environments?

Response: We agree with this and have incorporated your suggestions throughout the manuscript. A short line from line 325 to 331 been added in section 5. The phrase as per below:

“Due to certain applications of carbon fiber composites which exposed to high surrounding temperature (such as gun barrel, bridges, retrofitting of metallic building), seawater exposure (as base material for construction of yacht, canoe and boat) acidic and alkaline environment (used as reinforcement for damaged round steel pressure pipelines in which mostly used as economical transmission of liquids and gases), it is important to study the degradation of the material in this sector.”

Sincerely,

Norlin Nosbi

Round 2

Reviewer 2 Report

All the comments and questions has been discussed and addressed. The suggestions has been incorporated into the text.

I recommend to publish the manuscript in the revised form.